

# Bacterial meta-analysis of chicken cecal microbiota

Luis Alberto Chica Cardenas[1,2], Viviana Clavijo[3], Martha Vives[3] and Alejandro Reyes[1,2,4]

[1] Research Group on Computational Biology and Microbial Ecology, Department of Biological Sciences, Universidad de Los Andes, Bogotá, Colombia
[2] Max Planck Tandem Group in Computational Biology, Universidad de Los Andes, Bogotá, Colombia
[3] Centro de Investigaciones Microbiológicas, Department of Biological Sciences, Universidad de Los Andes, Bogotá, Colombia
[4] The Edison Family Center for Genome Sciences and Systems Biology, Washington University School of Medicine, St. Louis, MO, USA

Corresponding author
Alejandro Reyes,
a.reyes@uniandes.edu.co

## ABSTRACT

Poultry production is an industry that generates 90,000 metric tons of chicken meat worldwide. Thus, optimizing chicken growth and sustainable production is of great importance. A central factor determining not only production parameters, but also stability of the immune system and chicken health, is the diversity and variability of the microbiota present throughout the gastrointestinal tract. To date, several studies have investigated the relationship between bacterial communities and the gut microbiome, with limited data to compare. This study aims to create a bacterial meta-analysis based on studies using amplicon sequencing with Illumina sequencing technologies in order to build a baseline for comparison in future analyses of the cecal bacterial composition in chicken. A systematic literature review was performed (SYRF ID: e84f0468-e418-4eec-9da4-b517f1b4809d. Full project URL: https://app.syrf.org.uk/projects/e84f0468-e418-4eec-9da4-b517f1b4809d/detail). From all the available and analyzed manuscripts only nine contained full raw-sequence data available and the corresponding metadata. A total of 324 samples, comprising three different regions within the *16S rRNA* gene, were analyzed. Due to the heterogeneity of the data, each region was analyzed independently and an effort for a joint analysis was performed as well. Taxonomic profiling revealed 11 phyla, with *Firmicutes* as the most prevalent phylum, followed by *Bacteroidetes* and *Proteobacteria*. At genus level, 109 genera were found. Shannon metric for alpha diversity showed that factors like type of chickens (Commercial or experimental) and *16S rRNA* gene subregion have negligible effect on diversity. Despite the large number of parameters that were taken into account, the identification of common bacteria showed five genera to be common for all sets in at least 50% of the samples. These genera are highly associated to cellulose degradation and short chain fatty acids synthesis. In general, it was possible to identify some commonalities in the bacterial cecal microbial community despite the extensive variability and factors differing from one study to another.

## INTRODUCTION

Chickens are considered to be one of the main sources of food production worldwide (*Godfray et al., 2010*). Chicken meat production accounts for more than 128 million tons per year worldwide (*USDA Foreign Agricultural Service, 2019*). It is also considered as the most efficient production of animal protein, due to its property of doubling the weight of food ingested into meat weight at the end of the production cycle. Furthermore, chicken meat has been reported as a source of highly digestible proteins, minerals and vitamins with low levels of saturated fatty acids, which accompanied with a balanced diet, might help reduce the risk of having cardiovascular and endocrine problems (*Marangoni et al., 2015*). Likewise, the efficiency of poultry meat production has had an impact on the price of the product, which make it more accessible for a broader number of social classes than other sources of meat protein (*Scanes, 2007*).

The microbial community (microbiota) present in the gastrointestinal tract (gut) has been widely associated with factors involving the health of chickens such as the immune system, the physiology of the digestive system and exclusion of pathogens, as well as the performance in production (*Clavijo & Flórez, 2017*). Commensal bacteria have been positively associated with the generation and regulation of the mucus layer, which is involved in the protection of epithelial cells against pathogenic bacteria (*Clavijo & Flórez, 2017*). For instance, chicken gut microbiota might have a direct effect on controlling the prevalence of pathogenic bacteria by competitive exclusion (*La Ragione & Woodward, 2003*). Additionally, commensal bacterial metabolites, such as short chain fatty acids (SCFA), are responsible for the expansion of the absorption surface in the gut by increasing the number of its proliferating cells (*Kien et al., 2007*).

Among the different segments that comprise the chicken's gut, the cecum is the place where the food is retained the longest time. It presents the highest rate of water absorption and bacterial diversity (*Xiao et al., 2016*; *Shaufi et al., 2015*). Of all phyla that colonize the cecum, *Firmicutes* and *Bacteroidetes* are reported to be the most abundant. Their prevalence has been associated to their capacity for digesting cellulose and non-starch polysaccharides, which cannot be digested in the small intestine and leads to SCFA production (*Clavijo & Flórez, 2017*; *Sergeant et al., 2014*). Moreover, the *Firmicutes* that colonize the cecal region are also responsible for nitrogen cycling, which is highly associated with the efficiency of chickens to extract energy from food (*Mancabelli et al., 2016*; *Oakley et al., 2014*). Hence, most of the information available on chicken gut microbiota has being focused on this organ.

For microbial diversity analyses based on *16S rRNA* gene data it has been proven that the use of different sequencing technologies may alter species richness and estimates of microbial diversity for the same sample (*Allali et al., 2017*). Therefore, comparisons between samples sequenced by different methods or bacterial consensus using data mixtures might result in significant biases for further analysis. However, a bacterial consensus or meta-analysis might provide useful information in order to evaluate the effect of different factors in the normal composition of cecal microbiota. To the best of our knowledge, three meta-analysis of chicken cecal composition has been performed

(*Wei, Morrison & Yu, 2013*; *Waite & Taylor, 2014*; *Zou, Sharif & Parkinson, 2018*). Even though these studies were proposed as a good model for new projects, the data used was retrieved from samples sequenced by Sanger technology (*Wei, Morrison & Yu, 2013*), 454 pyrosequencing (*Waite & Taylor, 2014*) or by a mixture of data from different methods (*Zou, Sharif & Parkinson, 2018*). Nowadays, Illumina Sequencing platforms are the predominant methods to obtain biological sequence information. In consideration of the biases that a comparison of reads obtained by different sequencing strategies (*Plummer et al., 2015*), we performed a bacterial meta-analysis based on sequences generated with Illumina technology with the aim of identifying bacteria, that despite the variations associated with each study, are prevalent in the cecum and therefore, could be crucial in chicken gut modulation.

## MATERIALS AND METHODS

### Data collection and quality filters

In order to retrieve the data, an extensive literature search using "cecal microbiota composition in chickens" as search term was performed. Moreover, keywords such as "cecum", "broiler chicken", "microbiota composition", "16S rRNA" and "Illumina" were also taken into account. Different search engines (PubMed, Google Scholar, Elsevier) were used to conduct the search. Literature search was performed exclusively for english written papers and conducted until February 25, 2020. For the systematic search developed in this study, authors Luis Alberto Chica and Alejandro Reyes serve as screeners of the evaluated studies, whereas Martha Vives took the place of referee. However, there were no disagreements between screeners. A reverse search strategy, applying the same search terms described above, was made in SRA database in order to find data that might have been excluded in the initial search. Review articles were also checked for additional studies.

With the aim of choosing the most relevant studies, the following filters were used as inclusion parameters: (i) data must be available in public databases, (ii) metadata must be provided within the manuscript or in the database where the data was published and (iii) the number of samples reported in the article must coincide with the data submitted to the database. Those three steps are encompassed within a two-step exclusion criteria according to PRISMA guidelines (*Moher et al., 2009*), in which the screen criteria encompasses all studies that did not have available data and the eligibility criteria states for all studies in which the metadata was absent or the number of samples reported was different to the data submitted in public databases (Fig. 1). The entire set of *16S rRNA* gene sequences used for this analysis was retrieved from the Sequence Read Archive (SRA) database (*Leinonen, Sugawara & Shumway, 2010*) and downloaded using the SRA Toolkit software (*Leinonen, Sugawara & Shumway, 2010*). Trimmomatic software (*Bolger, Lohse & Usadel, 2014*) was used to filter low quality reads using a sliding window of 4 and a minimum Phred score of 20. Due to the lack of reported information regarding barcodes, primers and adapters for the different samples, a headcrop of 15 bp was performed for all samples. In order to validate trimming procedures, we conducted a primer search in raw and clean reads by using the specific primers reported in each study and the software Seqkit v0.14 (*Shen et al., 2016*) (Table S1).
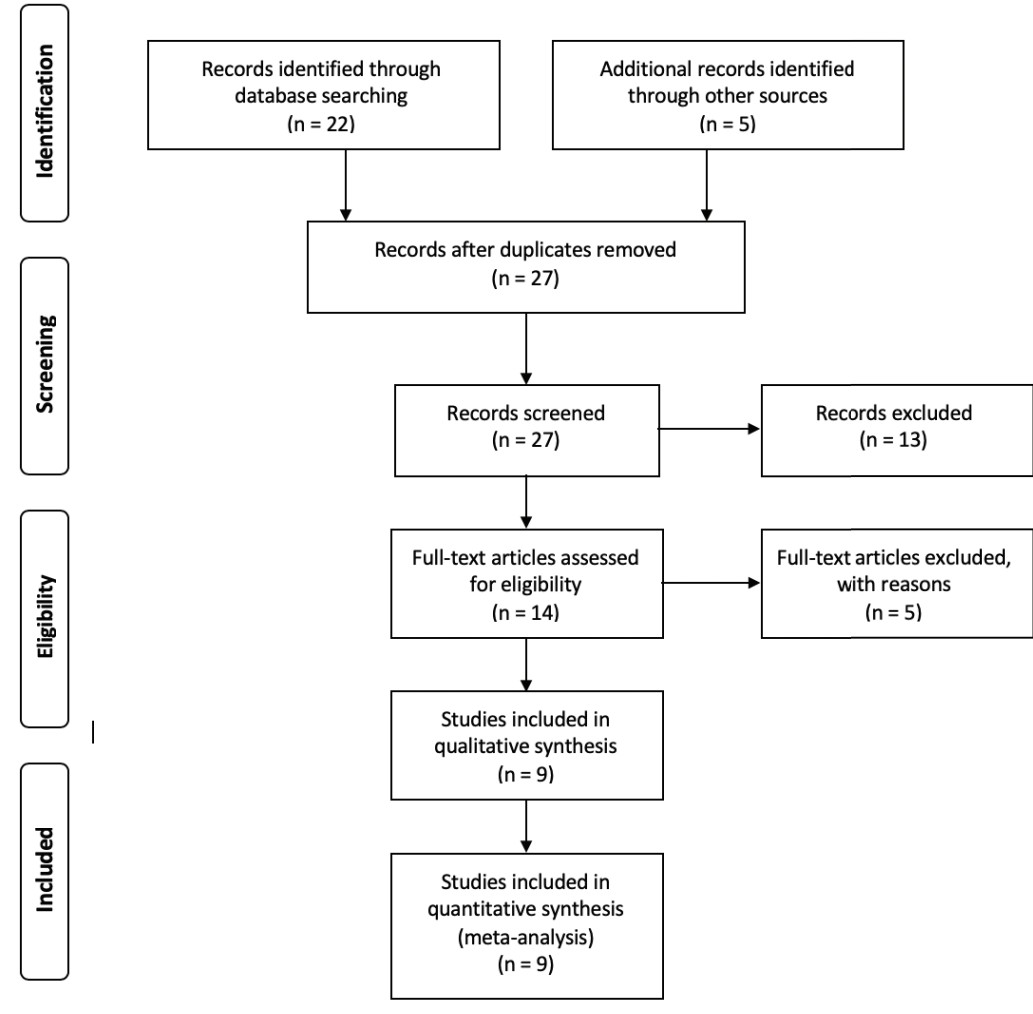

**Figure 1 Flow chart showing the filtering steps, resulting in the nine articles that were used for this meta-analysis.** This flow chart was made according to the Preferred Reporting Items for Systematic Reviews and Meta-Analyses (PRISMA) (*Moher et al., 2009*). Four steps are considered to be crucial for the correct evaluation of different records, in order to include them into the analysis. Whereas identification step only involves the finding of records and the elimination of redundant papers, in the screening step, all records in which data was not available are removed. After the screening process, the eligibility step involves the exclusion of all records that, even with available data, were removed for multiple reasons. For the purpose of this article, those reasons are considered as filters ii and iii (see Methods: "Data Collection and Quality Filters"). Finally, the inclusion step accounts for the final number of records used for the analysis.

## ASVs assignment

Due to the diversity of the hypervariable regions sequenced by the different studies, we split the entire dataset in three subsets. Each subset comprises samples generated with varying primers and corresponding to different hypervariable regions (V3, V4, V3V4). Deblur software (*Amir et al., 2017*) implemented in Qiime2 (*Bolyen et al., 2019*) was used for performing denoising and dereplication of the reads by applying a truncation length of 130 bp. ASV tables generated by deblur were filtered by a minimum frequency of 10, where singletons and doubletons were also excluded. In order to compare the taxa

identified by the different amplicon regions, ASV tables were summarized at the genus level using sklearn classifier for taxonomic classification. GreenGenes database v13.18 (*DeSantis et al., 2006*) was used for training the classifier. Subsequently, all taxa that did not reach genus level were removed from collapsed tables. Finally, filtered tables resulting from the three datasets were merged together.

## Core microbiome generation, taxonomic evaluation and diversity estimates

Independent core microbiome analysis was performed for all ASVs tables that were collapsed at genus level. Two approaches to define the core microbiome were assessed. (i) a strict approach closer to the general definition of core, where a genus should be present in at least 80% of the samples and (ii) a more relaxed search requiring a genus to be present in more than 50% of the samples. By using an in-house script, Venn diagrams were constructed with the aim of relating common genera present on the three hypervariable regions. Moreover, a core microbiome analysis per study was performed, selecting genera that were present in at least 80% of the samples of each given study. A comparison of the study specific core microbiomes was then performed to identify taxa that were present in the nine cores microbiome. The merged ASV table was transformed into relative frequency for computing taxonomic heatmaps at phylum and genus levels. For diversity estimates, rarefaction was performed at a sequence depth of 500. Alpha diversity was estimated using Shannon index and Observed species metrics. Bray-Curtis metric was employed to calculate Beta diversity. The significance of alfa diversity differences was evaluated by a Kruskall Wallis test. Meanwhile, the influence of different hypervariable regions on beta diversity estimates, was evaluated by a permutational analysis of variance (PERMANOVA), both tests as implemented in Qiime2.

## RESULTS

The literature search resulted in 27 studies that contained information fulfilling the search criteria (Table 1). After applying the filters described previously (see "Materials and Methods"), nine studies were selected to perform this meta-analysis (Fig. 1). Those studies contained a total of 324 samples and 36,743,185 of single end sequences, with a number of sequences per sample varying between 15,181 and 1,548,296. After applying sequencing quality filters, the number of sequences was reduced to 8,843,573. Among the different hypervariable regions, 4,178,126 sequences covered the V4 *16S rRNA* gene region, 1,897,781 the V3 gene and 3,280,185 covered both the V3 and V4 region (V3V4) (Table S2).

After denoising the reads, filtering the resulting ASV tables, collapsing all taxa at genus level and merging the ASVs, 109 genera and 11 phyla were identified. *Firmicutes* was the most prevalent phylum, followed by *Bacteroidetes* and *Proteobacteria*. In some cases, a given taxonomic profile was highly related to a single study, however, a similar abundance profile of *Firmicutes* and *Proteobacteria* was seen in more than 95% of the samples (Fig. 2A). At genus level *Oscilospira*, *Bacteroides, Helicobacter* and *Lactobacillus* were found as the most representative genera (Fig. 2B).

**Table 1 Characteristics of the manuscripts reviewed to perform the analyses.**

| Study | Available in public databases | Complete metadata | Information in databases according to article | Used for this study | Index |
|---|---|---|---|---|---|
| Mancabelli et al. (2016) | x | x | x | x | 1 |
| Shaufi et al. (2015) | x | x | x | x | 2 |
| Ballou et al. (2016) | x | x | x | x | 3 |
| Costa et al. (2017) | x | x | x | x | 4 |
| Xu et al. (2016) | x | x | x | x | 5 |
| Xia et al. (2019) | x | x | x | x | 6 |
| Varmuzova et al. (2015) | x | x | x | x | 7 |
| Biasato et al. (2018) | x | x | x | x | 8 |
| Zhou et al. (2016) | x | x | x | x | 9 |
| Awad et al. (2016) | x | | | | NA |
| Han et al. (2016) | x | | | | NA |
| Kim et al. (2018) | | | | | NA |
| Kollarcikova et al. (2019) | | | | | NA |
| Krueger et al. (2017) | x | | | | NA |
| Li et al. (2016) | | x | | | NA |
| Ma et al. (2017) | | | | | NA |
| Marimuthu et al. (2019) | | | | | NA |
| Mon et al. (2015) | | x | | | NA |
| Park, Lee & Ricke (2016) | | | | | NA |
| Park et al. (2017) | | | | | NA |
| Polansky et al. (2016) | | | | | NA |
| Saxena et al. (2016) | | | | | NA |
| Varmuzova et al. (2016) | | | | | NA |
| Wang et al. (2018) | x | | | | NA |
| Wu et al. (2019) | | | | | NA |
| Xiao et al. (2016) | | | | | NA |
| Yan et al. (2017) | | | | | NA |

Note:
Each column represents one of the search parameters that were taken into account in order to select the most suitable studies. Articles that fulfilled each criterion were labeled with an X. The column titled "Used for this study" lists the studies selected for the analysis, whereas the last column contains the index assigned to each study, which is used as reference in different figures. NA refers to Not Applicable.

For the evaluation of common bacterial taxa, core microbiome of the samples was assessed by two different approaches. Common bacteria in (i) at least 50% and (ii) at least 80% of the samples were evaluated. The traditional cut-off of 80% abundance for core definition showed *Oscillospira* as the only genus that was identified in all 3 regions (Fig. 3A). After relaxing the threshold and evaluating the 50% core microbiome at genus level, 5 genera were found to be common for all sets (*Oscillospira, Lactobacillus, Faecalibacterium, Clostridium* and *Ruminococcus*). In addition, 6 genera were shared by V3 and V4 regions, 3 were exclusively shared between V3 and V3V4 and 2 genera were common for V4 and V3V4 regions (Fig. 3B). It is important to highlight that all genera that were exclusively found in a particular region were present in all studies

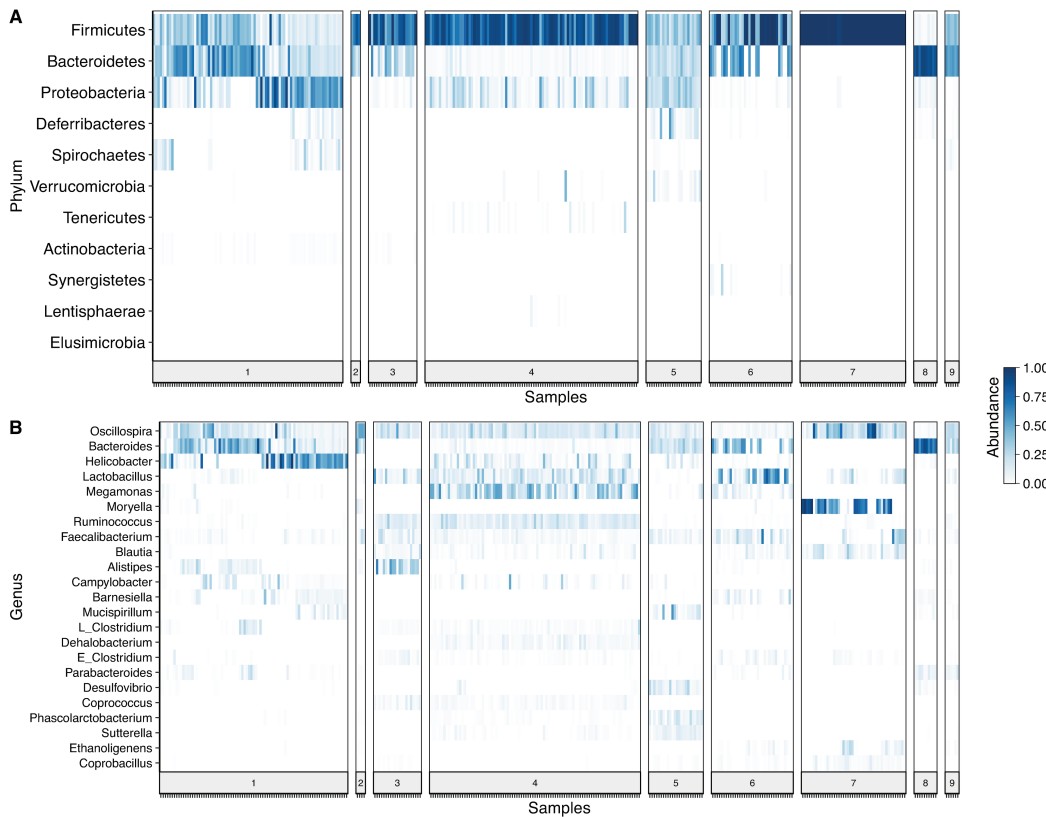

**Figure 2 Heat-map showing relative abundance of each sample from each study at different taxonomic assignments.** (A) Phylum and (B) genus level, for the merged ASV table. The most prevalent genera are shown (accounting for up to 90% of cumulative abundance). Index numbers at the bottom represent each study that was analyzed (Table 1). Relation of index numbers and metadata for each study is also shown on Table S1. In the *y*-axis, "L_Clostridium" refers to Clostridium genus assigned to Lachnospiraceae family and "E_Clostridium" refers to Clostridium genus assigned to Erysipelotrichaceae family.

concerning that specific region, but not necessarily in all samples (Table S3). Finally, we evaluated the core microbiome per study, here, *Oscillospira* remained as the only genus shared by all the studies. Interestingly, multiple taxa were part of the core microbiome of several but one or two studies. Overall, *Oscillospira, Faecalibacterium, Lactobacillus, Clostridium, Bacteroides, Blautia, Ruminococcus* and *Coprobacillus* were present in the core of 6 studies or more (Fig. 3C; Table S4), which includes all genera identified at the 50% overall core-microbiome.

Due to the high number of metadata categories considered in this analysis, alpha diversity variation was evaluated based on four categories: (i) *16S rRNA* gene hypervariable region (ii) study, (iii) type of chickens (chickens under commercial rearing conditions or chickens rearing under controlled environment "Experimental") and (iv) chicken breed. Variables such as age, due to the number of studies with missing data make it impossible to consider it as a variable for comparison. Alpha rarefaction plot for all samples using a depth of 500 sequences per sample showed that all samples reached a saturation asymptote, which confirms that the samples can be used for further analyses and
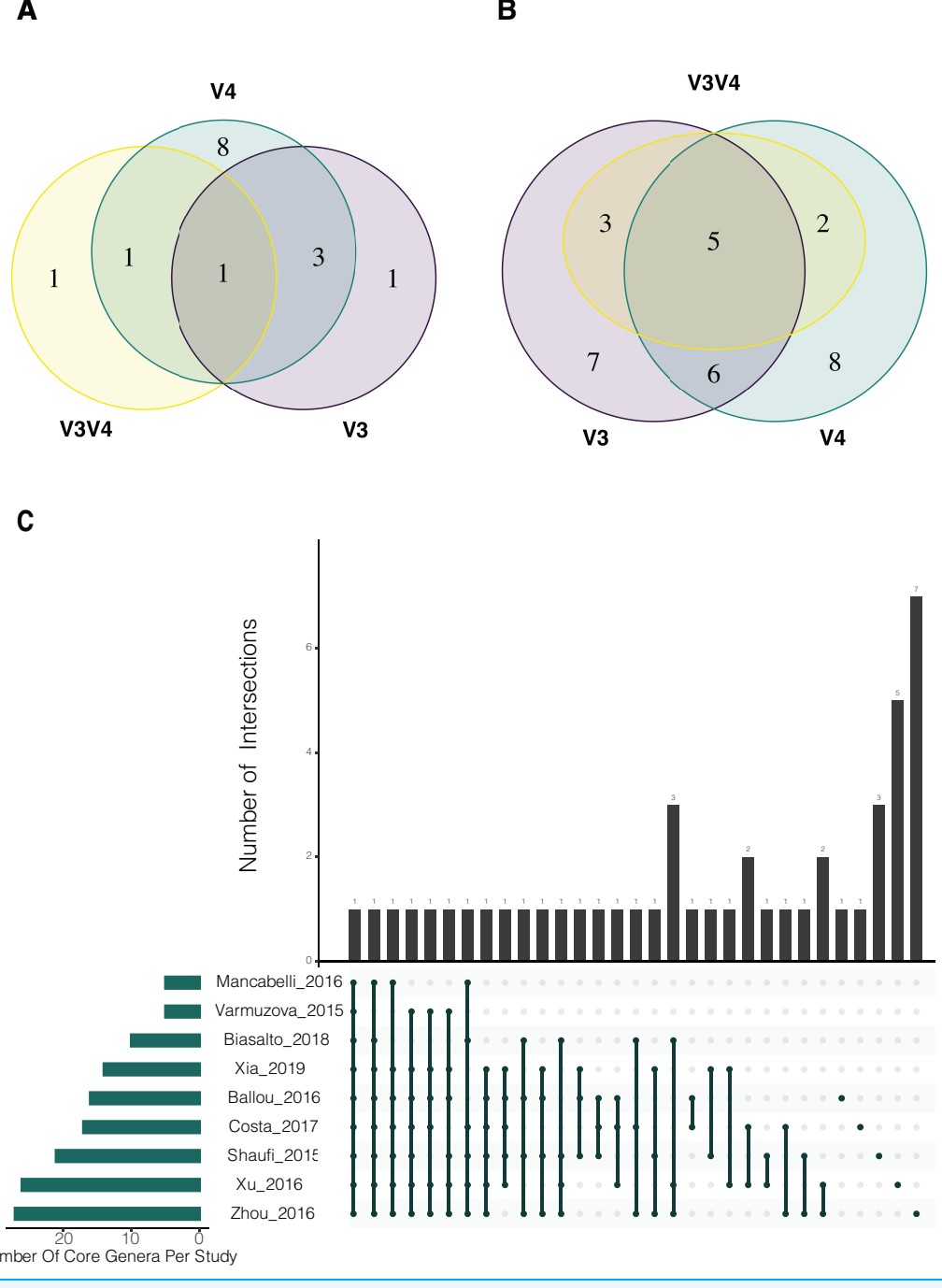

**Figure 3 Analysis of shared taxons in the three different hypervariable regions considered in the study.** (A) Analysis at genus level, using a 80% threshold, which means that bacteria must be present in at least the 80% of the samples. (B) Genus level at 50% threshold. (C) Core microbiome per study using an 80% threshold. Number of Core genera per study are shown on the horizontal bar plot, while the number of genera shared in the core of one or more studies are indicated in the vertical bar plots, the shared studies for each specific columns are shown by the connected dots.

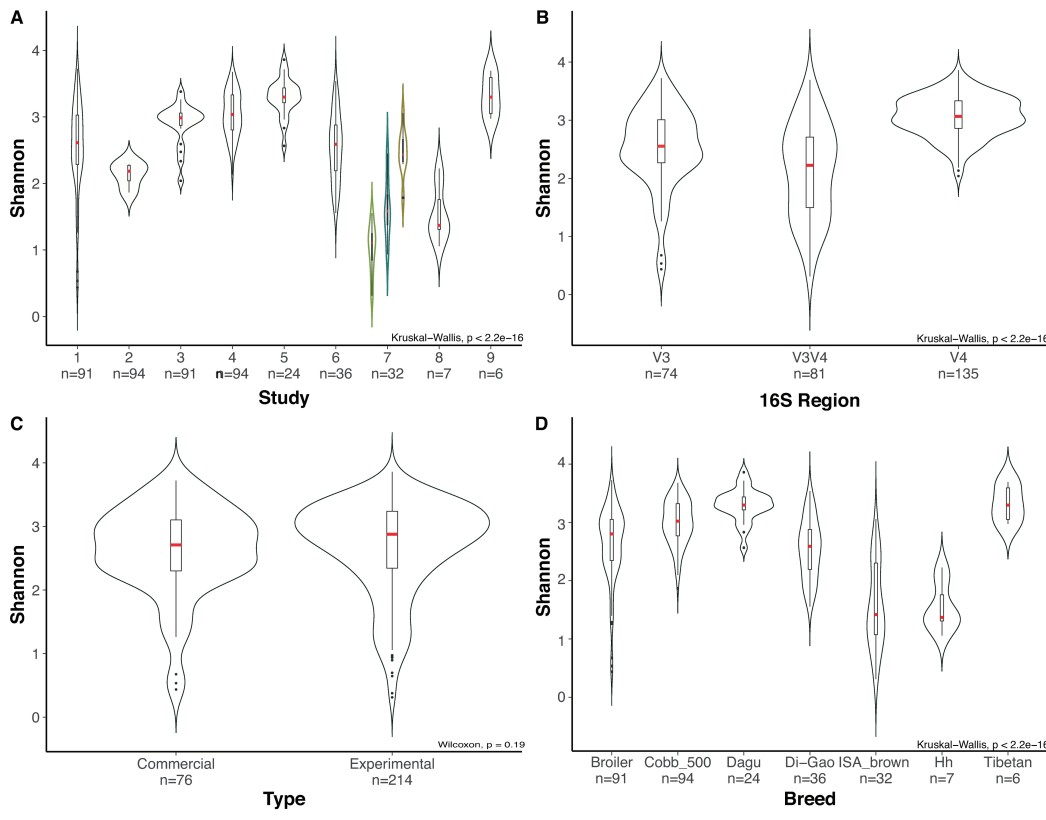

**Figure 4 Evaluation of alpha diversity of the entire set of samples by using Shannon metric and Kruskal–Wallis statistics for significance.** Violin plots represent different groups for each category. (A) Study. (B) Hypervariable region amplified from the *16S rRNA* gene. (C) Type of chickens (commercial, experimental). (D) Breed of the chickens, where Hh corresponds to Hubbard Hybrid breed and Broiler to all fattening breeds that are not specified. Boxplots represent the interquartile range and in red the median for each group. The number of samples in each group is shown right below the name. For Panel A, Indexes of metadata were used for visualization purposes and are related with the specific studies in Table S1. The colored violin plots correspond to *Varmuzova et al. (2015)* study, which is highlighted due to the inclusion of Salmonella *enteriditis* samples (green) that showed lower diversity than treated samples with plant extract (blue) and control samples (yellow) from the same study.

confirming a relatively low diversity in the chicken cecum (Fig. S1). Rarefaction process discarded 33 samples that did not fulfill the minimum sampling depth, leaving 290 samples for the diversity analyses.

Most variation for alpha diversity was observed when comparing among the different studies. Interestingly, one of the groups of samples that presented low diversity values was a subset of the study of *Varmuzova et al. (2015)*, which contains chickens infected with *Salmonella enteriditis*, showing a significant reduction (Wilcoxon, *p*-value = 0.009), compared with healthy chickens of the same study and healthy chickens from other studies (Fig. 4A). Likewise, when analyzing the effect of the selected hypervariable region, we noticed a significant difference on Shannon values for all groups (Kruskal Wallis, *p*-value = 1.9e−19), being V4 the region with the higher values (Fig. 4B). In order to prove if the difference presented in the V3V4 group was a consequence of the infected chickens

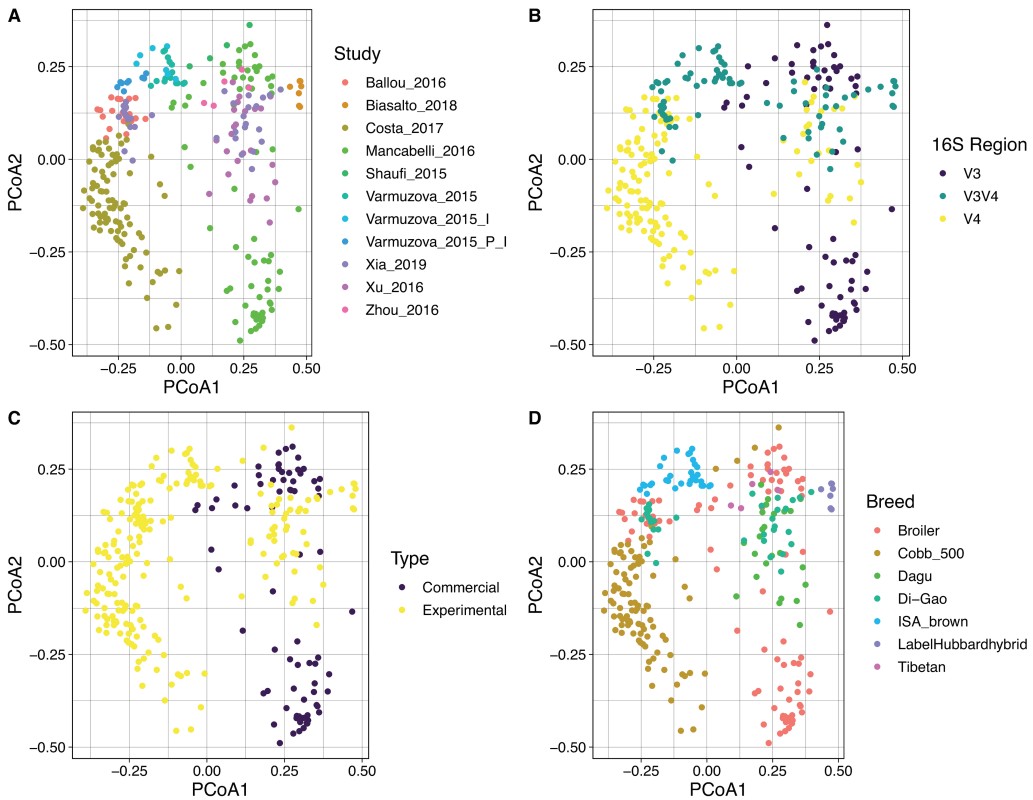

**Figure 5 PCoA showing the spatial dispersion of the samples. Distance matrix was constructed based on Bray–Curtis metric.** Coordinate plots represent (A) Study; the study of *Varmuzova et al. (2015)* was split due to the high intra-study variation obtained. Varmuzova_2015 = Control Chickens and uninfected Chickens treated with the plant extracts; Varmuzova_2015_I = Infected chickens; Varmuzo-va_2015_P_I = Infected chickens, treated with the plant extracts. (B) Hypervariable region of the *16S rRNA* gene. (C) Type of chickens (commercial, experimental), when commercial chickens refer to animals raised on normal production environments and experimental refers to animals that were kept into environments with controlled conditions. (D) Breed of the animals, where Broiler corresponds to all fattening breeds that are not specified.

that was previously described, we removed those samples from the dataset and performed again the analysis obtaining similar results. The comparison of filtered V3V4 group against V4 (Wilcoxon, $p$-value = 1.2e−13) and V3 (Wilcoxon, $p$-value = 0.0067) groups independently, still showed significant differences on both cases. On the other hand, comparisons between commercial and experimental chickens reveal no significant variation (Wilcoxon, $p$-value = 0.188) (Fig. 4C). When using the breed of the chickens as variable, even though significant differences were observed, we cannot be sure of its relevance due to the lack of repeating breeds over different studies (Fig. 4D).

By using Bray Curtis metric, distance matrices were built, and beta diversity was estimated for the same categories evaluated above. No strong clustering was observed by any of the variables analyzed (Fig. 5), however, some recognizable clustering was observed for the samples belonging to *Mancabelli et al. (2016)* and *Costa et al. (2017)*. Further statistical analysis using Permanova pairwise method was applied on the variable Study and hypervariable region, both analyses showed significant differences. Whereas all

pairwise comparisons performed to the variable study reported $q$-values ranging between 0.00125 and 0.423, being the comparison of two groups of the same study (control and infected chickens) the only with no significant differences, the comparisons applied to the hypervariable region variable report all $q$-values of 0.001.

## DISCUSSION

The aim of this study was to create a baseline for future research in order to give a higher insight into the general composition of the microbial communities in the cecum of chickens. This analysis aimed to collect the largest possible dataset of studies describing bacterial cecal content using Illumina technologies and amplicon sequencing.

Unfortunately, the study was limited since many of the published studies did not have their data available in public databases or did not share the metadata related with each project, making impossible to use the data or evaluate and replicate the results. Problems such as the lack of data available and the high number of sequences that were excluded after the quality filters were applied can be avoided by implementing common quality standards in the experimental design, data generation and data deposition. Only with serious commitment to high standards it is possible to achieve reproducible and comparable results among different studies. Another important point to be taken into account is the lack of a consensus on the specific hypervariable region of the *16S rRNA* gene to focus. In order to compare the different results obtained it was needed to collapse the ASVs tables at genus level and merge them.

After taxonomic classification of the three sets of samples (different hypervariable regions) at phylum and genus level, the results showed the predominance of *Firmicutes* and *Bacteroidetes* in most of the samples. These results are concordant with previously reported consensus based on sequences generated by different sequencing platforms (*Wei, Morrison & Yu, 2013*). When looking at the genus level, the number of genera reported were much higher in this meta-analysis than what was reported for individual studies (*Xu et al., 2016*; *Zhou et al., 2016*; *Biasato et al., 2018*) and comparable with the meta-analysis of Sanger generated sequences, performed by *Wei, Morrison & Yu (2013)*, in which 117 genera were found to be present in the chicken gastrointestinal tract. Thus, the increase in sensitivity, achieved by the higher number of sequences generated using Illumina sequencing technology, might lead to comparable results, even when fewer samples were used.

The behavior of diversity estimates for *16S rRNA* gene regions showed that the V4 region presents the highest diversity values at ASV level when comparing with V3 and V3V4. This behavior is supported by experimental results (*Sperling et al., 2017*). On the other hand, when the V3V4 dataset is separated in groups of control chickens and chickens infected with *Salmonella enteriditis*, control chickens show the same diversity behavior than animals belonging to the other studies, however differences when comparing V3V4 region against V3 and V4 regions were still present. Those differences are likely due to factors such as biases on the primer's specificity, although the V3V4 primers used on the different studies did not have the same sequence, or the resolution obtained by the

inter-specific variability of the amplified region. When comparing diversity estimates between commercial and experimental chickens no difference was found, leading to assume that variables like temperature, food sources, water supply, housing, hygiene, and the number of animals per cage may not have as significant effects as other variables on bacterial diversity, although it is important to highlight that only two studies encompass commercial samples, from which a single study groups 92% of the samples, leading to a potential bias. Despite the large number of variables inherent to each study, the collapse of ASVs tables to genus level allowed to diminish the influence of those variables in beta diversity analysis and led to the clusterization of several samples corresponding to different studies. Although, neither of the metadata categories considered in this metanalysis could group the samples, leading to assume that beta diversity grouping is driven by multiple categories.

The evaluation of common bacteria, in different percentages of the samples at the genus level, revealed the presence of several bacterial genera with an important role in food conversion. The genera *Faecalibacterium*, *Oscillospira*, *Lactobacillus*, Clostridium and *Ruminococcus* were found to be present in more than 50% of the samples. The importance of *Faecalibacterium* has been attributed to their capacity to express acetyl-CoA acetyltransferase and several enzymes involved in the production of butyrate (*Polansky et al., 2016*). Production of butyrate could also be accomplished by *Clostridum* and *Oscillospira* species and due to the role of butyrate on anti-inflammatory responses and grow performance, the remarkable prevalence of this genera could be explained (*Biasato et al., 2018*). On the other hand, *Ruminococcus* and *Lactobacillus* have been extensively described as major colonizers of cecum and ileum, and as part of *Ruminococcaceae* family, their role in SCFA production has been proposed (*Wang, Lilburn & Yu, 2016*; *Rinttilä & Apajalahti, 2013*; *Wei, Morrison & Yu, 2013*; *Gophna, Konikoff & Nielsen, 2017*). Moreover, the ability of *Ruminococcus* to degrade cellulose, can explain its major presence and abundance in the cecum (*Devillard et al., 2004*).

Beta diversity results using different studies as variable showed no clear differentiation of samples from the same study, only two studies showed this behavior. These findings differ with the analysis performed by *Zou, Sharif & Parkinson (2018)*, in which the clustering of samples from the same study is strong. Is not clear the potential source of the variations in the results observed, but they might come from the different efforts in cleaning and normalizing the datasets in order to make them as comparable as possible thus reducing the biases inherent to experimental procedures. However, differences in the software used for the analysis and other technical factors could also have an influence in this variation.

Even though this study may have potential biases related to the data availability and different experimental procedures performed by different groups in different countries, it presents a useful model of comparison and baseline for future studies when cecal microbiome composition wants to be assessed (Table S5; Additional File 1). Despite the heterogeneity of the samples, several bacterial genera, with implications in food conversions rate, were reported to be present in an important proportion of the sample.

## CONCLUSIONS

Microbial communities' meta-analysis is a very valuable tool to identify commonalities as well as sources of variation that structures the communities in different environmental settings. Our study showed that the first and most important limitation is the availability of the data and thorough and complete description of the metadata associated. Once those factors were taken into consideration, the source of variation can be both technical and experimental, such as the choice of primers to use or the presence of certain pathogens as part of the experimental design; however, no single factor was the major driver of inter-sample variation, although in some cases samples from a given study clearly clustered together.

Finding commonalities among the different datasets depends on the taxonomical level of resolution desired, at phylum level we were able to detect 11 different phyla and a general structure with dominance of Firmicutes, Bacteroidetes and Proteobacteria. At higher taxonomical levels such as genus, although some genera were identified in a majority (>50%) of the samples overall or within studies, few of them were consistently present suggesting a functional redundancy among closely related bacteria. Most of these common bacteria have been associated with growth and health of different hosts. As we improve our methods for generating and reporting metagenomic and metabarcoding studies, more insights into the ecological and functional role of the different members of bacterial communities associated to a given host will be elucidated.

## ACKNOWLEDGEMENTS

The authors would like to acknowledge doctors Maryam Chaib, Angela Viviana Peña and Juan Manuel Anzola for their help in the revision of the manuscript. Likewise, we thank the IT Services Department and ExaCore - IT Core-facility of the Vice Presidency for Research & Creation at the Universidad de Los Andes that allow us to perform the computational analysis.

### Funding

The authors received no funding for this work.

### Competing Interests

Martha Vives is an Academic Editor for PeerJ.

### Author Contributions

- Luis Alberto Chica Cardenas conceived and designed the experiments, performed the experiments, analyzed the data, prepared figures and/or tables, authored or reviewed drafts of the paper, and approved the final draft.
- Viviana Clavijo conceived and designed the experiments, analyzed the data, authored or reviewed drafts of the paper, and approved the final draft.

- Martha Vives conceived and designed the experiments, authored or reviewed drafts of the paper, and approved the final draft.
- Alejandro Reyes conceived and designed the experiments, analyzed the data, prepared figures and/or tables, authored or reviewed drafts of the paper, and approved the final draft.

## Data Availability

We didn't develop or generated any raw data or code for the current study, all raw data used was previously published and is referenced and the software used is third party and has already been published.

## Supplemental Information

Supplemental information for this article can be found online at http://dx.doi.org/10.7717/peerj.10571#supplemental-information.

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
