# Peer review of "Bacterial meta-analysis of chicken cecal microbiota"

_PeerJ, doi:10.7717/peerj.10571_

## Round 0.1 · original submission · Major Revisions

The reviewers suggest some modifications, both technical and conceptual, that seem reasonable and not very difficult to apply.

·

Basic reporting

The manuscript is written in a good English with the sufficient amount of background information. It has the professional structure and self-contained.

Experimental design

The research question is well defined. The paper is providing a meta-analysis of taxonomic diversity of cecal bacterial taxonomy diversity in chicken. Methods are well-described and appropriate expect some minor issues listed below.

Validity of the findings

The findings look valid and meaningful. Although the papers with raw data are provided it would be good to add to the Table S1 SRA IDs of the raw read datasets and PubMed IDs of the papers to make the search easier. Additionally, it would be beneficial for reproducibility to provide exact commands for Trimmomatic and QIIME2 used for the data processing.

Additional comments

Line 135: The title should be “Core microbiome”
Line 189 and below: “Salmonella Enteriditis” Usually species name of a bacteria is written in italics and lower case.
Line 199: The text states that Kruskal Wallis test was implemented, but Figure 4C shows Wilcoxon test.
Line 207: “(p-value = 0.001)” QIIMEII should give you q-value which is more appropriate for pairwise PERMANOVA than unadjusted p-value. Was p-value the same for all comparisons?
Line 437: Please provide explanation of the two-step exclusion criteria on the Figure 1 as in the methods section they are described as a one step with 3 (i-iii) criteria. It would be good to provide more explanation of the steps in this flow chart.
Line 444: It is impossible to associate index numbers to the study from the table 1 as according to the table S1 the numbers are associated to studies even not in alphabetical order. Please add indexes to the table 1.
Line 455: Please clarify meaning of asterisks. Also is not clear what base mean stands for. Kruskal Wallis test shows if different groups are drawn from the same distribution or not. Was the test implemented as a pairwise against some artificial mean group? What does a different number of asterisks mean? Maybe authors should consider remove the asterisks as I don’t see if they were discussed.

Reviewer 2 ·

Basic reporting

Please see detailed comments below.

Experimental design

Please see detailed comments below.

Validity of the findings

Please see detailed comments below.

Additional comments

The ms by Cardenas et al. “Bacterial meta-analysis of chicken cecal microbiota” synthesizes and reanalyzes 16S rRNA sequence data from nine previous studies with a goal to identify a core set of microbial taxa in the chicken cecal microbiome. While this is an interesting objective, the ms needs to be strengthened considerably before it would be suitable for publication in my opinion. Some general and specific comments are below with some suggestions for improvement.


General comments:
Similar work has recently been performed that the authors do not mention. Please see and discuss: NPJ Biofilms Microbiomes 4, 27 (2018).

Searching for a universal set of core taxa in the chicken cecal microbiome is a valid and important question. The main result of five ‘core’ genera is interesting but needs to be strengthened in several ways. This result is such that it can only be defined by loosening criteria to allow only 50% membership. This is not surprising given that the data analyzed comes from such different contexts, arguably undermining this result.

As Figure 5 shows, there are significant differences among communities according to breed, study, experimental group, etc. These results are interesting and have been shown previously. One additional parameter not shown here that is also well known to have a significant effect on the cecal/GI microbiome is the age of the birds. Given the heterogeneous population of birds collected from the nine studies in question, it’s no wonder that finding a core set of taxa requires such non-stringent criteria. I agree with the value of searching for a core microbiome, but the ms as it currently stands is trying to force a square peg into a round hole. Recognizing that things are different and then trying to show how they are the same is somewhat illogical but rather than being a fatal flaw in this ms, this situation presents some interesting opportunities that can easily improve the ms significantly.

Because of the community-wide differences among studies (many of which are significant as the authors show with permanova), it would be more logical and informative to stratify the data by study, grouping into categories as appropriate. Identifying sets of core taxa for each study/condition would be more meaningful and biologically relevant. These core sets of taxa could then be compared to identify commonalities AND differences. This type of approach would significantly strengthen the work and increase its value to researchers working in this field.

Finally, the number and types of ASVs within each core genus should be reported. Collapsing the ASV classifications to the genus level is valid, but since this information exists, it should be reported. These data would be helpful to other researchers and should be reported for full disclosure as the authors advocate. Reporting these in a phylogenetic context (i.e. tree) would be most helpful.



Some additional specific comments:

L105-107: unconventional reporting and not meaningful description of roles, rubrics, decision-criteria, etc.

L109: review

L119-120 – Seems like a kind of crude approach. How do you know this was sufficient? Many barcode/adapter combinations are up much longer than this and are of variable length. Either search for specific sequences and/or identify sequences that were removed. Since ASVs are being derived, these trimming procedures can have important effects on the results.

L129: taxa

L131: Please note Greengenes does not appear to have been updated since 2013.
Also see BMC Genomics volume 18, 114 (2017) for comparisons of 16S rRNA databases.


L229: Need specific citation for reference here.
L231-2: This argument needs to be refined. Please cite evidence that 99% or 97% OTUs can merge genera. Additionally, this argument doesn’t seem to be valid since ASVs were only used to identify genera.

---

## Round 0.2 · accepted · Accept

The revievers' comments have been addressed.

·

Basic reporting

The authors improved manuscript sufficiently for publication.

Experimental design

no comment

Validity of the findings

no comment

Additional comments

no comment